# Self-recognition drives the preferential accumulation of promiscuous CD4+ T-cells in aged mice

Neha R Deshpande[1,2], Heather L Parrish[1], Michael S Kuhns[1,2,3]*

[1]Department of Immunobiology, University of Arizona College of Medicine, Tucson, United States; [2]Arizona Center on Aging, University of Arizona College of Medicine, Tucson, United States; [3]BIO-5 Institute, University of Arizona College of Medicine, Tucson, United States

**Abstract** T-cell recognition of self and foreign peptide antigens presented in major histocompatibility complex molecules (pMHC) is essential for life-long immunity. How the ability of the CD4+ T-cell compartment to bind self- and foreign-pMHC changes over the lifespan remains a fundamental aspect of T-cell biology that is largely unexplored. We report that, while old mice (18–22 months) contain fewer CD4+ T-cells compared with adults (8–12 weeks), those that remain have a higher intrinsic affinity for self-pMHC, as measured by CD5 expression. Old mice also have more cells that bind individual or multiple distinct foreign-pMHCs, and the fold increase in pMHC-binding populations is directly related to their CD5 levels. These data demonstrate that the CD4+ T-cell compartment preferentially accumulates promiscuous constituents with age as a consequence of higher affinity T-cell receptor interactions with self-pMHC.

*For correspondence: mkuhns@email.arizona.edu

**Competing interests:** The authors declare that no competing interests exist.

## Introduction

Each T-cell expresses a T-cell receptor (TCR) encoded by rearranged gene segments and non-germline nucleotides. Estimates of TCR diversity imply a repertoire that can bind a universe of self and foreign peptides embedded within self-major histocompatibility complex molecules (pMHC) (*Davis and Bjorkman, 1988*). Yet, this potential cannot be realized. Thymic development limits clonal representation to T-cells bearing TCRs within an affinity window for self-pMHC (*Savage and Davis, 2001*; *Yin et al., 2012*; *Klein et al., 2014*), while peripheral space physically constrains the number of T-cells present to recognize foreign-pMHC (*Mason, 1998*; *Vrisekoop et al., 2014*). Finally, time—with its age-associated changes in thymic expression of tissue-restricted antigens (TRAs), thymic architecture, antigen experience, and homeostasis—imposes an overarching pressure that limits the binding capacity of a repertoire for self- and foreign-pMHC to each constituent's prior history of TCR–pMHC interactions (*Nikolich-Zugich, 2008*; *Surh and Sprent, 2008*; *Chinn et al., 2012*; *Griffith et al., 2012*). How these pressures shape the capacity of the CD4+ T-cell compartment to bind pMHC over the lifespan remains largely unexplored.

Aging is associated with increased susceptibility to infections and decreased responsiveness to vaccines, suggesting that individual repertoires converge on a point where their diversity is insufficient to bind and/or mount a protective response to foreign-pMHC (*Vazquez-Boland et al., 2001*; *Nichol, 2008*; *Nikolich-Zugich, 2008*). Consistent with this idea, TCR diversity within both the CD4+ and CD8+ T-cell compartments contract from adult to old mice in parallel with thymic involution (*Ahmed et al., 2009*; *Rudd et al., 2011*; *Britanova et al., 2014*), and the number of CD8+ T-cells that bind distinct foreign class I pMHC in unprimed mice decreases over the lifespan (*Yager et al., 2008*; *Rudd et al., 2011*; *Decman et al., 2012*; *Smithey et al., 2012*). Here, we explored how aging impacts the number

**eLife digest** The immune system's T cells help the body to recognize and destroy harmful pathogens, such as viruses and bacteria. T cells 'remember' immunity-inducing fragments, called antigens, from the pathogens they have encountered. This memory then allows the immune system to quickly fend off infections if those pathogens, or even related pathogens, invade again. Vaccines exploit the ability to form immunological memory by exposing the body to harmless forms of the pathogen, or even just particular antigens from it. This allows the T cells to learn how to identify the pathogen without any risk of illness.

Vaccines have been extremely successful and have helped to virtually eliminate some diseases. However, for reasons that are unclear, the immune systems of older adults become less functional, so vaccines often lose their effectiveness. Paradoxically, as people age T cells become more likely to attack the body's cells, causing autoimmune diseases like arthritis. Understanding what happens to aging T cells to cause these immune changes may help scientists design vaccines that remain effective as people age.

Little is known about what happens to a particular type of T cell—the CD4+ T cells—as people age, even though this population plays a critical role in providing other immune cells with detailed instructions on when and how to fight a pathogen. Now, Deshpande et al. show that CD4+ T cells undergo a remarkable set of changes in aging mice. Mice that are nearing the end of their natural lifespan have fewer CD4+ T cells than younger mice. However, those CD4+ T cells that remain are more likely than CD4+ T cells from younger mice to be able to recognize multiple antigens. This increase in the proportion of multitasking CD4+ T cells corresponds with an increased tendency of these cells to bind to the body's own cells. If similar changes occur in older people, this may help explain some age-related autoimmune diseases. Yet, the relationship between the increase in multitasking CD4+ T cells and the decrease in immune function with aging remains to be fully explored.

The challenge for scientists now is to determine how these age-related changes in CD4+ T cells affect immune responses to vaccines or pathogens in older individuals. One implication of this work is that CD4+ T cell responses may be too robust and out of balance with other arms of the immune system. This could even lead to conditions such as autoimmunity. Alternatively, while there may be more CD4+ T cells that can multitask by recognizing multiple antigens, their ability to respond appropriately to infections or vaccinations may be diminished. What is clear from the work of Deshpande et al. is that the rules that have been defined for immunity in adults change with aging. The rules that govern immunity in the elderly must be more clearly defined to realize the goal of designing immunotherapies, such as vaccines, that provide protection throughout the lifespan.

of naive and memory phenotype CD4+ T-cells available to bind pMHC, their relative affinity for self-pMHC, and their capacity to bind foreign-pMHC. We report that, while the absolute number of CD4+ T-cells decreases over time, those that remain have an increased affinity for self-pMHC and an increased capacity to bind foreign-pMHC.

## Results

Unprimed old (18–22 months) C57BL/6 mice were found to have fewer CD4+ T-cells in their secondary lymphoid organs than adults (8–12 weeks) due to a loss of naive (CD44lo) T-cells, as expected given thymic involution (*Figure 1A,B*) (*den Braber et al., 2012*). The number of memory phenotype (CD44hi) CD4+ T-cells increased with aging (*Figure 1C*). This could be due to prior antigen experience and/or homeostatic proliferation (*Nikolich-Zugich, 2008*; *Surh and Sprent, 2008*).

To assess steady-state TCR engagement, we measured CD5 expression, as a surrogate for the strength of tonic TCR–pMHC interactions (*Azzam et al., 1998*; *Smith et al., 2001*; *Mandl et al., 2012, 2013*; *Persaud et al., 2014*; *Vrisekoop et al., 2014*; *Fulton et al., 2015*); CD3 levels, which decrease upon TCR engagement (*Valitutti et al., 1995*); and BrdU incorporation to assess proliferation in unprimed mice. CD5 was higher on memory CD4+ T-cells in adult mice relative to adult naive T-cells, as expected (*Mandl et al., 2013*), while both naive and memory CD4+ T-cells in old mice had higher CD5 expression relative to adult naive cells (*Figure 1D*). An inverse relationship was observed

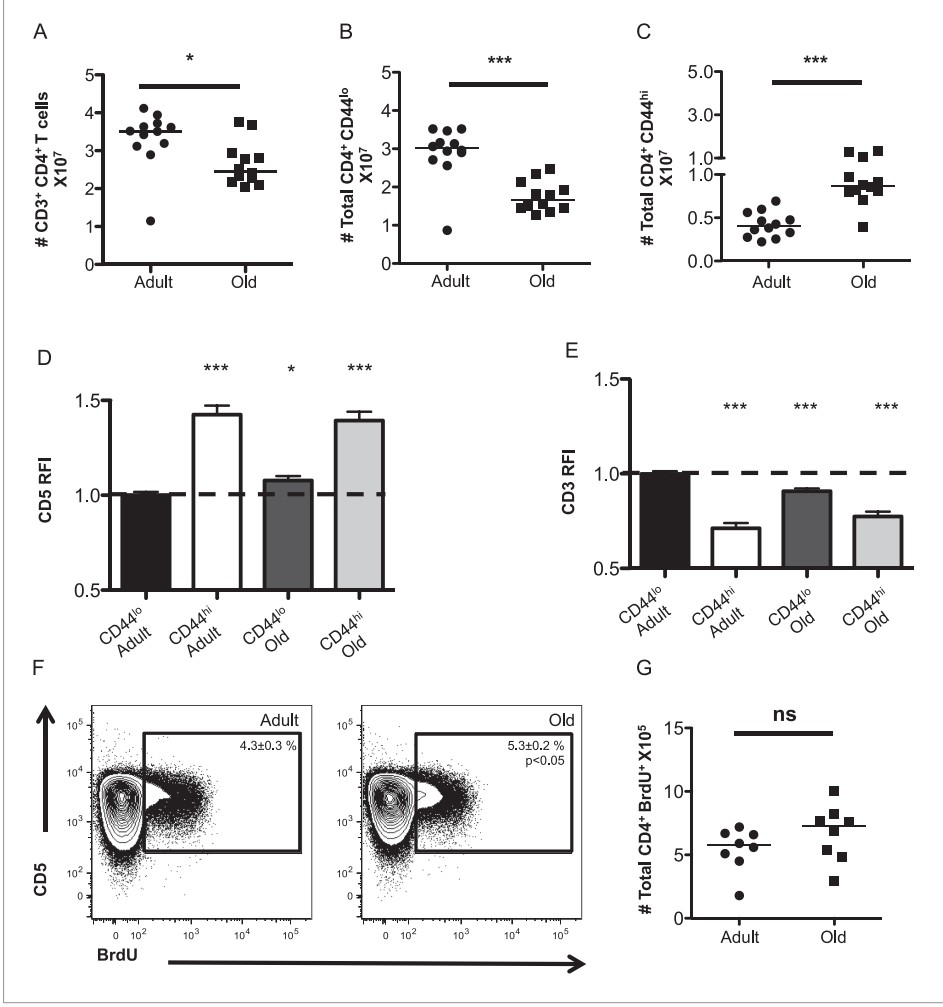

**Figure 1**. The CD4+ T-cell compartment contracts but accumulates CD44hiCD5hi cells with aging. The absolute numbers of T-cells in unprimed adult (8–12 weeks) and old (18–22 months) mice are shown as (**A**) total CD4+ T-cells in secondary lymphoid organs, (**B**) CD4+ CD44lo (naïve) T-cells and (**C**) CD4+ CD44hi (memory phenotype) T-cells. Data are concatenated from three experiments, 4 mice/group. Horizontal bar indicates median (*p < 0.05, ***p < 0.0001; Mann–Whitney). (**D**) Relative fluorescent intensity (RFI) of CD5 expression on adult and old CD44hi and CD44lo CD4+ T-cells relative to CD5 expression on adult CD44lo CD4+ T-cells (dotted line). Data represent four experiments with 4 mice/group (***p < 0.0001, *p < 0.05; Mann–Whitney). (**E**) RFI of CD3 expression on adult and old CD44hi and CD44lo CD4+ T-cells relative to CD3 expression on adult CD44lo CD4+ T-cells (dotted line) (***p < 0.0001; Mann–Whitney). Results represent seven experiments with 4 mice/group. (**F**) Concatenated contour plots (4 mice) showing CD5 vs BrdU incorporation in unprimed adult and old total CD4+ T-cells. Percent BrdU+ of total CD4+ T-cells ± SEM is shown in the inset (*p < 0.05 Mann–Whitney adult compared to old). (**G**) Absolute numbers of CD4+ BrdU+ T-cells. Results are representative of two experiments with 4 mice/group.

between CD5 and CD3 levels, consistent with CD5 reflecting tonic TCR engagement (*Figure 1E*). Finally, cells with high CD5 expression incorporated the most BrdU in adult and old mice, consistent with tonic TCR interactions driving homeostatic proliferation (*Figure 1F*). A higher frequency of BrdU+ cells was observed in old mice compared with adults. However, since the total number of CD4+ T-cell drops in old mice this did not result in significantly more BrdU+CD4+ T-cells (*Figure 1F,G*). Altogether, the data indicate that the CD4+ T-cell compartment increases in clonal representation of constituents with higher intrinsic affinity for self-pMHC.

Age-related changes in the capacity of the CD4+ T-cell compartment to bind foreign-pMHC were evaluated via tetramer enrichment (all class II pMHC tetramer validation is shown in

*Figure 2—figure supplements 1, 2*). I-A$^b$ tetramers presenting an immunodominant peptide (aa 641–655) from West Nile Virus (WNV) envelope protein (E641:I-A$^b$) were used because WNV lethality increases over the lifespan of mice and humans, making it a useful model for investigating age-related defects in susceptibility to viral infection and vaccine efficacy (*Brien et al., 2008*, *2009*; *Uhrlaub et al., 2011*; *Suthar et al., 2013*). Two-color tetramer enrichment (*Nelson et al., 2015*) revealed more cells binding E641:I-A$^b$ in old mice than adults (*Figure 2—figure supplement 3*).

To determine if this is unique to E641:I-A$^b$, we also enumerated CD4$^+$ T-cells with distinct recognition properties by using a tetramer made with a subdominant ovalbumin peptide (326–338) in I-A$^b$ (OVA:I-A$^b$), and an allogeneic tetramer made with the moth cytochrome c peptide (88–103) bound to I-E$^k$ (MCC:I-E$^k$) (*Savage et al., 1999*; *Malherbe et al., 2004*; *Moon et al., 2007*; *Brien et al., 2008*). OVA:I-A$^b$ was considered to be subdominant because immunization with OVA elicited a smaller response than E641 in isolation and failed to mount a response upon co-immunization with E641 (*Figure 2—figure supplements 2, 4*). OVA:I-A$^b$ monomer is also less SDS-stable than E641:I-A$^b$ at room temperature (not shown), and pMHC stability is directly related to immunodominance (*Lazarski et al., 2005*). Alloreactive cells were enumerated because they are likely to be selected on a broader range of self-pMHC and represent a broader subset of the CD4$^+$ T-cell compartment (*Felix and Allen, 2007*; *Chu et al., 2009*).

CD4$^+$ T-cells bound to E641:I-A$^b$, OVA:I-A$^b$, and MCC:I-E$^k$ were simultaneously enriched from individual animals using anti-His beads against the 6× His-tag on the alpha and beta subunits of each pMHC (*Figure 2A–F* and *Figure 2—figure supplement 5*). This yielded more E641-bound adult cells than the anti-PE/APC beads (*Figure 2G* and *Figure 2—figure supplement 3E*). Since tetramers cannot detect all CD4$^+$ T-cells that respond to a given class II pMHC via weak TCR–pMHC interactions (*Sabatino et al., 2011*), the more avid His-tag enrichment is likely to detect T-cells that bind tetramers with lower avidity.

More naive and memory cells were observed to bind a single pMHC specificity in old mice compared with adults when using dump tetramer gating (*Figure 2G–I* and *Figure 2—figure supplements 5, 6*) (*Savage et al., 1999*). This indicates that the increase in CD4$^+$ T-cells binding E641:I-A$^b$ is not unique. Rather, since CD4$^+$ T-cells decline with aging, those that are left appear to bind foreign pMHC more promiscuously. Consistent with this interpretation, the number of naive cells binding OVA+MCC was higher in old mice compared with adults, as were the number of memory cells binding E641+OVA or OVA+MCC (*Figure 2J–L* and *Figure 2—figure supplements 7–9*). Altogether, these data provide evidence that the CD4$^+$ T-cell compartment becomes polyspecific over time.

Such results could reflect age-related changes in thymic selection, homeostatic signals, or both. To evaluate the former, we enriched thymocytes from adult and old mice with E641:I-A$^b$, OVA:I-A$^b$, and MCC:I-E$^k$ tetramers. The frequency of E641-bound CD4 single positive (SP) cells was higher for old thymocytes compared with the adults, while the frequency of OVA and MCC-bound CD4SPs did not differ (*Figure 3* and *Figure 3—figure supplement 1*). CD4SPs binding two distinct tetramers were not detected amongst the small number of tetramer-enriched samples. This is not surprising given that dual binders average <10% of a peripheral population (*Figure 2—figure supplement 6*). Since thymic output remains constant as a function of size over time (*Hale et al., 2006*), the higher frequency of E641-bound CD4SP thymocytes in old mice suggests that more E641-binders leave the thymus of old mice than adults on a daily basis. However, mature CD4$^+$ T-cells re-entering the thymus increase from ~10% in adult mice to ~20% in old mice (*Hale et al., 2006*). Our analysis cannot resolve CD4SPs from mature CD4$^+$ T-cells, so the impact of recirculation on our analysis is unclear. Nevertheless, the data suggest that age-related changes in thymic selection impact the clonal representation and binding capacity of the CD4$^+$ T-cell compartment.

Finally, we investigated how tonic TCR engagement relates to the capacity of the CD4$^+$ T-cell compartment to bind foreign-pMHC (*Mandl et al., 2013*). CD5 levels on the tetramer-bound adult populations, relative to those on the total adult CD4$^+$ T-cell population, directly correlated with the fold increase in the absolute number of these populations over time (*Figure 4A,B*). Steady-state BrdU incorporation for adult and old tetramer-bound CD4$^+$ T-cells also mirrored the rank order (OVA>E641>MCC) of CD5 expression seen in both the naive and memory populations (*Figure 4C,D*). Thus, CD5 levels are predictive of the fold-increase in pMHC-specific CD4$^+$ T-cell subsets with aging, suggesting a link between affinity for self-pMHC, homeostatic proliferation, and expansion over time.

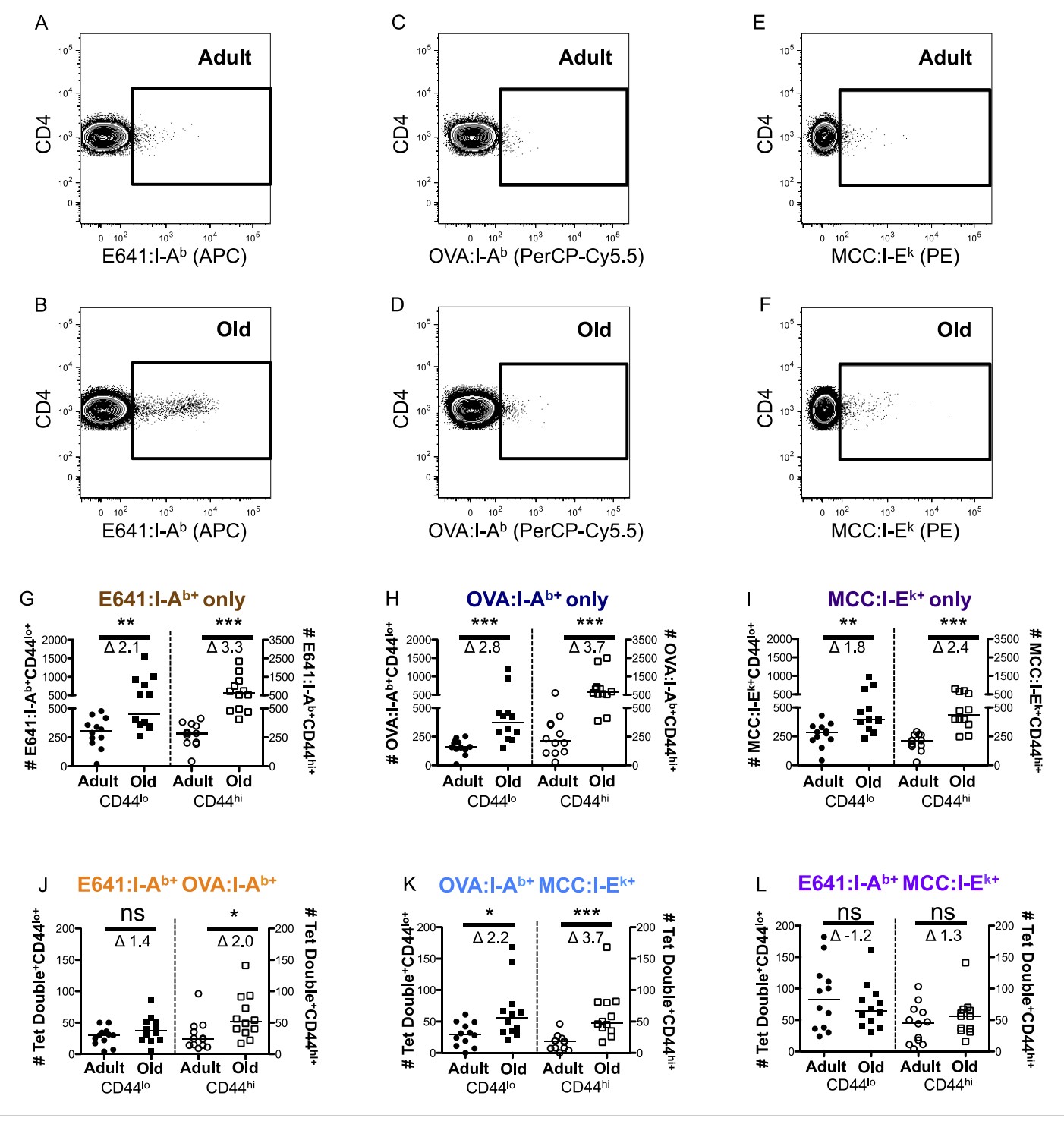

**Figure 2**. CD44$^{lo}$ and CD44$^{hi}$ CD4$^+$ T-cells binding immunodominant, subdominant, and allogeneic pMHC increase with time. Representative plots of CD4$^+$ T-cells bound to (**A** and **B**) E641:I-A$^b$, (**C** and **D**) OVA:I-A$^b$, and (**E** and **F**) MCC:I-E$^k$ tetramers in adult (top) and old (bottom) mice. Absolute number of CD4$^+$ CD44$^{lo}$ (left Y-axis) and CD4$^+$ CD44$^{hi}$ (right Y-axis) T-cells bound to (**G**) E641:I-A$^b$, (**H**) OVA:I-A$^b$, or (**I**) MCC:I-E$^k$ tetramers only enumerated after dump tetramer analysis ('Materials and methods'), or those binding (**J**) E641:I-A$^b$ + OVA:I-A$^b$, (**K**) OVA:I-A$^b$ + MCC:I-E$^k$, or (**L**) E64:I-A$^b$ + MCC:I-E$^k$ tetramers in combination enumerated after both dump and two-color tetramer analysis ('Materials and methods'). Bars indicate median (*p < 0.05, **p < 0.005, ***p < 0.0001, ns = non-significant; Mann–Whitney). Fold change (Δ) in means between adult and old is shown. Results are from three experiments with 4 mice/group.

The following figure supplements are available for figure 2:

*Figure 2. continued on next page*

*Figure 2. Continued*

**Figure supplement 1**. Tetramer validation on T-cell hybridomas.

**Figure supplement 2**. Tetramer validation for in vivo primed CD4[+] T-cells.

**Figure supplement 3**. WNV-specific CD4[+] T-cells increase over the lifespan.

**Figure supplement 4**. E641 is immunodominant to OVA.

**Figure supplement 5**. Gating scheme for identification of tetramer[+] cells.

**Figure supplement 6**. Poly-specific cells form a very small fraction of a particular total tetramer[+] population.

**Figure supplement 7**. Two-color analysis of E641+OVA polyspecific cells ± dump tetramer exclusion.

**Figure supplement 8**. Two-color analysis of OVA+MCC polyspecific cells ± dump tetramer exclusion.

**Figure supplement 9**. Two-color analysis of E641+MCC polyspecific cells ± dump tetramer exclusion.

## Discussion

Advances in the analysis of clonal representation, pMHC-binding capacity, and functionality within the T-cell repertoire are contributing to a broader understanding of the rules that govern its composition and function. While most studies focus on adult mouse or human T-cells, when immunity is at its peak, there is a growing appreciation that the pressures imposed by time on thymic selection and peripheral space result in a repertoire that continuously evolves in each individual. Here, we contribute to our basic understanding of T-cell biology by reporting that the size of the CD4[+] T-cell compartment contracts with aging but, unlike CD8[+] T-cells, the capacity of CD4[+] T-cells to bind foreign-pMHC increases over the lifespan.

Thymic involution could contribute to these changes in multiple ways. A decrease in cortical thymic epithelial cells and changes in antigen processing could increase competition for positively selecting pMHC (*Chinn et al., 2012*; *Klein et al., 2014*), favoring higher TCR affinity for self-pMHC. In addition, decreased expression of TRAs on fewer medullary TECs (*Chinn et al., 2012*; *Griffith et al., 2012*) could lead to competition for negatively selecting pMHC with aging. Experimentally limiting thymic selection differentially impacts the CD4[+] and CD8[+] T-cell compartments, with CD4[+] T-cells becoming more polyspecific and CD8[+] T-cells becoming more pMHC focused (*Huseby et al., 2005*; *Chu et al., 2009*, *2010*; *Wang et al., 2009*; *Yin et al., 2012*). Thus, age-related changes in the thymus would be expected to restrict negative selection and result in a CD4[+] T-cell compartment with a broader binding capacity, as observed here. It is also noteworthy that T-cells can productively rearrange two TCRα subunits and express two TCRs that increase reactivity to self- and allo-pMHC (*Ni et al., 2014*). Whether T-cells expressing two TCRs increase over time remains unexplored.

Changes in peripheral space are also likely to contribute to the results reported here. A link between higher affinity for self-pMHC and residence within the CD4[+] T-cell memory pool of adult mice was previously reported (*Mandl et al., 2013*). Here, we extended this observation to naive and memory CD4[+] T-cells in old mice, indicating that affinity for self-pMHC influences clonal fitness over time. This would be akin to the affinity of TCR–pMHC interactions influencing clonal fitness within a polyclonal response to cognate antigens (*Lanzavecchia and Sallusto, 2002*; *Gett et al., 2003*; *Malherbe et al., 2004*). Indeed, CD5 levels on adult tetramer-binding memory subsets directly correlated with their fold expansion over the lifespan showing that CD5 levels have a clear predictive value when identifying populations with a long-term advantage for clonal representation within the CD4[+] T-cell compartment.

Altogether, the data presented here suggest a more complex relationship between CD4[+] T cells and immune senescence than has been reported for the CD8[+] T cells. While an increase in binding capacity may compensate for a decrease in total CD4[+] T cell numbers, the consequences of this increase remain unclear. Certainly, a population with a higher affinity for self-pMHC and broader

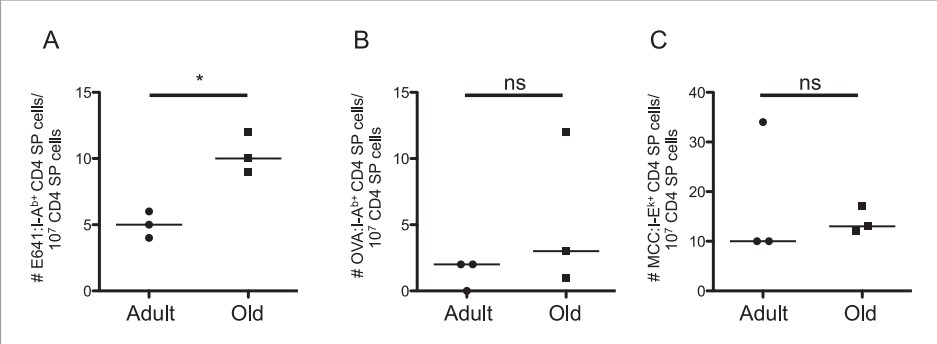

**Figure 3**. Evidence for changes in selection of E641-binding CD4SP thymocytes with aging. Frequencies of (**A**) E641: I-A$^{b+}$, (**B**) OVA:I-A$^{b+}$, and (**C**) MCC:I-E$^{k+}$ CD4 single positive (SP) thymocytes per 10$^7$ CD4SP thymocytes are shown. Horizontal bar indicates median (*p < 0.05 and ns = non-significant; Mann–Whitney). Each dot represents the results from 4–5 mice pooled/group as described in 'Materials and methods'.

The following figure supplement is available for figure 3:

**Figure supplement 1**. Dump tetramer plus two-color gating scheme for identification of tetramer$^+$ CD4SP thymocytes.

binding capacity poses obvious risks that could explain the increase in age-related autoimmune diseases, such as rheumatoid arthritis and giant cell arteritis (*Weyand et al., 2003*; *Mohan et al., 2011*). Coupling functional analysis with the results presented here will be important to gain a better understanding of the functionality of the CD4$^+$ T cell compartment over the lifespan.

## Materials and methods

### Mice
Old (18–22 months) male C57BL/6 mice were obtained from the National Institute of Aging breeding colony Bethesda, MD. Adult (8–12 weeks) male C57BL/6 mice were purchased from the Jackson Laboratory Bar Harbor, Maine. Mice were maintained under specific pathogen-free conditions in the animal facility at The University of Arizona. Experiments were conducted under guidelines and approval of the Institutional Animal Care and Use Committee of The University of Arizona.

### Peptides, CFA, and immunizations
Synthetic peptides Env 641–655 (E641: PVGRLVTVNPFVSVA) and OVA 323–339 (OVA: ISQAVHAA-HAEINEAGR) were purchased at >95% purity from 21st Century Biochemicals Marlborough, MA. Complete Freund's adjuvant (CFA) was purchased from Sigma–Aldrich St. Louis, MO. Mice were immunized with 50 µg peptide in 50 µl CFA on each side of the base of the tail.

### Tetramers
Class II pMHC monomers were generated with baculovirus expression vectors, based on pAcGP67A (BD Pharmingen San Jose, CA), encoding acidic or basic leucine zippers (generous gift of KC Garcia) according to the approach of Teyton and colleagues (*Scott et al., 1996*). The full extracellular domains of I-E$^k$ alpha and I-A$^b$ alpha were expressed as fusions with the acidic leucine zipper, a BirA acceptor peptide, and a 6× his tag. The full I-A$^b$ beta extracellular domain was expressed as fusions with the WNV Env 641–655 or OVA 326–338 peptides at the N-terminus, via a short linker similarly to Kappler and colleagues (*Crawford et al., 1998*), and at the C-terminus with the basic leucine zipper and a 6× his tag. I-E$^k$ beta fused to moth MCC 88–103 was otherwise the same.

Baculovirus stocks were made in Sf9 cells and large-scale protein production was performed in Hi5 cells as previously described (*Dukkipati et al., 2006*). pMHC complexes were purified from media by affinity chromatography using Ni-NTA affinity resin (Qiagen Valencia, CA) followed by biotinylation with BirA (Avidity, Aurora, CO.) and size exclusion chromatography with a Superdex-200 column (GE Healthcare Life

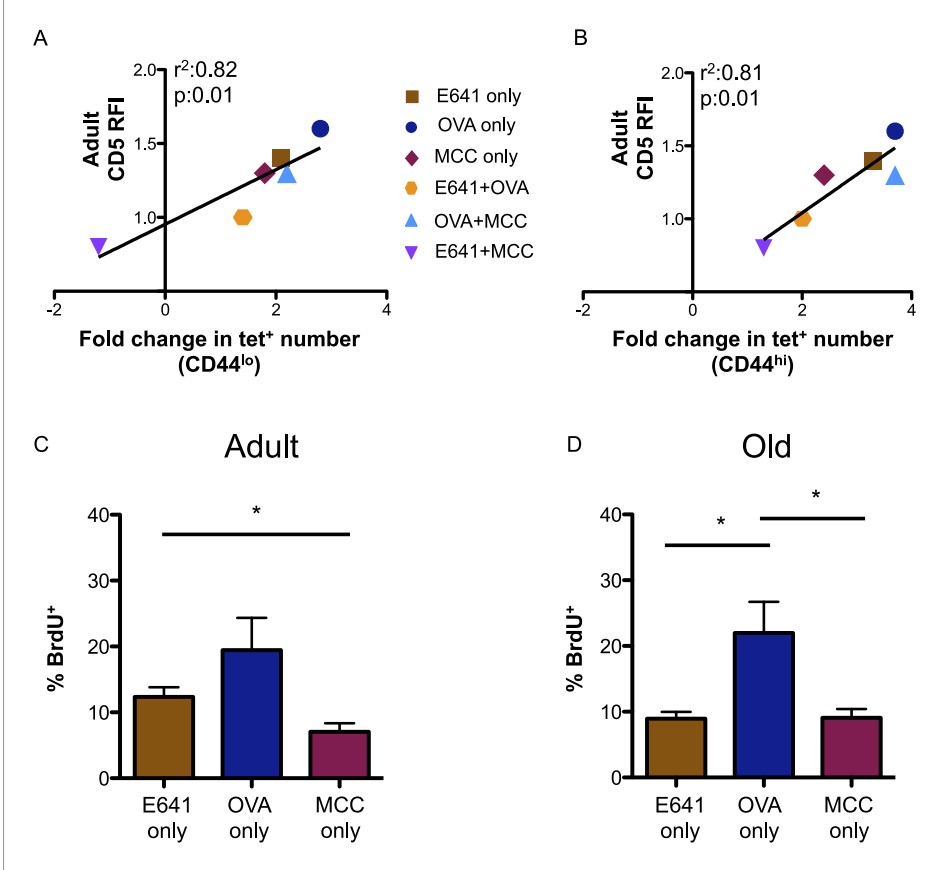

**Figure 4**. CD5 levels on adult CD4[+] T-cells correlate with expansion over time. Correlation between CD5 RFI for adult CD4[+] tetramer[+] T-cells and fold change in tetramer[+] cells between adult and old populations of (**A**) CD44[lo] and (**B**) CD44[hi] CD4[+] T-cells are shown as labeled. Linear regression was calculated using GraphPad Prism 5. Steady-state in vivo proliferation was assessed by measuring percent BrdU incorporation in tetramer single[+] (**C**) adult or (**D**) old CD4[+] T-cells derived from unprimed mice after 6 days of BrdU exposure (*$p < 0.05$; ANOVA followed by Dunn's post-test comparison). Results represent two experiments with 4 mice/group.

Sciences Pittsburgh, PA). Tetramers were created by mixing biotinylated peptide:I-A[b] or I-E[k] monomers with PE (Biolegend San Diego, CA)-, APC (Biolegend)-, or PerCPCy5.5 (eBiosciences San Diego, CA)-conjugated streptavidin at a molar ratio of 4:1 (Tetramer Concentration: 25 nM).

## pMHCII tetramer-based enrichment and analysis

Tetramer enrichment and analysis was performed as described previously (*Moon et al., 2007*) with slight modifications. Inguinal, cervical, axillary, popliteal, mesenteric, and lumbar lymph nodes were harvested along with the spleen from individual mice. Single-cell suspensions of lymph node and spleens were depleted of red blood cells with ACK lysis buffer (Gibco Life Technologies Grand Island, NY) and Fc blocked (mAb 2.4G2 hybridoma supernatant + 2% mouse serum [Caltag Laboratories Burlingame, CA], 2% rat serum [Jackson Immuno Research Laboratories, INC West Grove, PA]) on ice for 20 min. Each tetramer was added at a final concentration of 25 nM and incubated at room temperature in the dark for 1 hr. Cells were washed in FACS buffer (PBS + 2% FBS, 0.1% NaN₃) and resuspended in a final volume of 200 μl containing 25 μl of anti-PE and 25 μl of anti-APC microbeads (Miltenyi Biotec San Diego, CA) for two-color analysis of cells binding a single pMHC tetramers (*Stetson et al., 2002*; *Obar et al., 2008*; *Nelson et al., 2015*) or 50 μl of anti-His microbeads for simultaneous enrichment of cells binding three independent pMHC tetramers. After 30-min incubation at 4°C, cells were washed, resuspended in 3 ml FACS buffer and passed over a LS magnetic column at 4°C (Miltenyi Biotec) according to the manufacturer's instruction. The columns

were removed from the magnetic field and bound cells were eluted by allowing 4 ml of FACS buffer to pass through the column by gravity at 4°C. A second elution was performed by pushing 4 ml of FACS buffer through the column with a plunger at 4°C. The tetramer-enriched 'bound' fraction and an aliquot of flow-thru, or 'unbound' fraction, were stained with a cocktail of flourochrome-labeled antibodies for 30 min at 4°C (anti-CD19 [eBiosciences], anti-CD8α [eBiosciences], anti-CD11c [eBiosciences], anti-F4/80 [Biolegend], anti-CD3 [eBiosciences], anti-CD4 [eBiosciences], anti-CD44 [eBiosciences], anti-CD5 [BD Pharmingen]). Cells were washed, and the samples were analyzed with a LSRII cytometer (Beckton Dickinson Franklin Lakes, NJ). Analysis was performed using FlowJo software (Treestar Ashland, OR). Gating was performed as shown in figure supplements.

Tetramers are composed of pMHC monomers and streptavidin (SA) conjugated to a fluorescent protein (FP). Two-color tetramer enrichment and gating for a single pMHC specificity was performed as a method for reducing false-positives in tetramer analysis (*Stetson et al., 2002*; *Obar et al., 2008*; *Nelson et al., 2015*). The operating principle followed here is that cells which bind to tetramers via TCR–pMHC interactions will fall on a diagonal, since binding should be proportional for each, while those that bind to the FP in a TCR-independent manner should fall off the diagonal (*Figure 2—figure supplements 1, 3*). The dump tetramer approach was used for samples in which E641:I-A$^b$, OVA:I-A$^b$, and MCC:I-E$^k$ tetramers were used in a single sample for enrichment with anti-His beads (*Savage et al., 1999*; *Newell et al., 2009*). Here, cells bound to a tetramer of interest (e.g., E641:I-A$^b$) were gated and then those that also bound the other two tetramers (e.g., OVA:I-A$^b$ and MCC:I-E$^k$) were excluded as a 'dump' to enumerate cells bound to one tetramer species only (*Figure 2—figure supplement 5*). The dumped cells could be false-positives binding SA, MHC, or the dump tetramer-associated FP nonspecifically; however, they could be bound via TCR–pMHC interactions. Importantly, the operating principles of both the dump and two-color methods were employed to enumerate cells binding two tetramers at once. Specifically, to enumerate cells bound to a specific combination (e.g., E641:I-A$^b$ + OVA:I-A$^b$), those also binding the third tetramer (e.g., MCC:I-E$^k$) were dumped prior to enumerating cells bound to both tetramers of interest by two-color analysis. Combining the two approaches should enumerate cells binding tetramers via TCR–pMHC interactions and exclude dumped cells that bind SA, MHC, or the dump-associated FP and those excluded by two-color analysis that bind the FP associated with the tetramers of interest in a non-specific manner. Combining both approaches by sequential gating (*Figure 2—figure supplement 5*) yielded the numbers shown in *Figure 2J–L*. The same overall results were achieved if quadrant gating was used for two-color analysis after applying the dump gate (*Figure 2—figure supplements 7–9*).

## Thymocyte preparation

Thymi from old and adult mice were harvested in 1 ml of un-supplemented RPMI. Thymi from 4–5 old mice were pooled in order to take into account the drop in the total number of thymocytes in old mice due to thymic involution and to increase the total number of cells for the tetramer enrichment processing. Thymi were incubated with 3 ml of Accutase (eBiosciences) at 37°C for 30 min to achieve optimal cell detachment. Single cell suspension of thymocytes was depleted of red blood cells with ACK lysis buffer. The total number of thymocytes in old mice was 10-fold lower ($\sim$2 $\times$ 10$^7$) than adults ($\sim$2 $\times$ 10$^8$) due to thymic involution. The adult samples were then normalized for comparison by pooling 2 $\times$ 10$^7$ thymocytes from 4 to 5 adult mice. Thymocytes were Fc blocked on ice for 20 min. Cells were stained with E641:I-A$^b$–PerCP-Cy5.5, OVA:I-A$^b$–PE-Cy7, MCC:I-E$^k$–PE and each of these tetramers in a common FP (APC). Each tetramer was added at a final concentration of 25 nM. Tetramer enrichment was carried out as described above. The tetramer enriched 'bound' fraction and an aliquot of flow-thru, or 'unbound' fraction were stained with cocktail of flourochrome-labeled antibodies for 30 min at 4°C (anti-CD19, anti-CD8α, anti-CD11c, anti-F4/80, anti-CD3, anti-CD4, anti-CD5). Cells were washed and the samples were analyzed with a LSRII cytometer (Beckton Dickinson). Analysis was performed using FlowJo software (Treestar) as shown in *Figure 3—figure supplement 1*. The single color specificities of two of the tetramers (e.g., OVA and MCC) were used as dump tetramers prior to two-color analysis of the third tetramer (e.g., E641).

## Hybridoma cell lines

TCR negative 58α$^-$β$^-$ hybridomas cells were transduced with retroviral vectors encoding the OT II, 5c.c7 or 2B4 TCR, full-length CD3 subunits, and CD4 according to previously described protocols (*Kuhns and Davis, 2007*).

### In vivo proliferation assay

BrdU was administered to mice through drinking water at the concentration of 1 mg/ml + 1% glucose. Spleen and lymph nodes were harvested on day 6. Post-tetramer enrichment, cells were stained with cell surface antibodies (anti-CD3, anti-CD4, anti-CD19, anti-CD8α, anti-CD11c, anti-F4/80, anti-CD44, and anti-CD5) followed by intracellular anti-BrdU (BD Pharmingen) antibody according to BrdU flow kit protocol (BD Biosciences).

### Statistical analysis

Mean fluorescent intensity of cell surface antibodies and intra-cellular antibodies were obtained from FlowJo software (Treestar). Statistical analyses were performed using the Mann–Whitney t-test for non-parametric data, ANOVA followed by Dunn's post-test for multiple comparisons of non-parametric data or linear regression for analyzing correlation. All statistical analysis was performed using GraphPad Prism software.

## Acknowledgements

We thank Janko Nikolich-Zugich, Dominik Schenten, Megan Smithey, Jennifer Uhrlaub, Caleb Glassman, and Sing Sing Way for critical comment and feedback on the manuscript. We also thank Mark Lee for thoughtful comments and technical assistance, as well as members of the Frelinger, Wu and Schenten labs for critical discussions. MSK is a Pew Scholar in Biomedical Sciences, supported by the Pew Charitable Trusts. This work was supported by the BAA-NIAID-DAIT-NIHAI2010085 (MSK).

## Additional information

### Funding

| Funder | Grant reference | Author |
| --- | --- | --- |
| National Institutes of Health (NIH) | BAA-NIAID-DAIT-NIHAI2010085 | Michael S Kuhns |
| Pew Charitable Trusts | | Michael S Kuhns |

The funders had no role in study design, data collection and interpretation, or the decision to submit the work for publication.

### Author contributions

NRD, Conception and design, Acquisition of data, Analysis and interpretation of data, Drafting or revising the article, Contributed unpublished essential data or reagents; HLP, Acquisition of data, Contributed unpublished essential data or reagents; MSK, Conception and design, Analysis and interpretation of data, Drafting or revising the article, Contributed unpublished essential data or reagents

### Ethics

Animal experimentation: This study was performed in strict accordance with the recommendations in the Guide for the Care and Use of Laboratory Animals of the National Institutes of Health. All of the animals were handled according to approved institutional animal care and use committee (IACUC) protocols (#08-102) of the University of Arizona.

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
