## [Decision Letter]

Thank you for sending your work entitled “Affinity for self drives the preferential accumulation of promiscuous CD4^+^ T cells over the lifespan” for consideration at *eLife*. Your article has been favorably evaluated by Tadatsugu Taniguchi (Senior editor) and three reviewers, one of whom is a member of our Board of Reviewing Editors.

The Reviewing editor and the other reviewers discussed their comments before we reached this decision, and the Reviewing editor has assembled the following comments to help you prepare a revised submission.

1) The manuscript demonstrates that the specificity (and perhaps self-pMHC affinity) landscape of the TCR repertoire shows interesting aging-related changes in the CD4 T cell compartment that are distinct from those reported for the CD8 T cell compartment (12). The findings are correlative at this point with no data implicating any mechanisms by which these distinctions arise, and they require greater rigor in both data and interpretation.

2) The TCR specificity of the tetramer binding, the assay upon which all the results depend, must be established with greater rigor than at present. While the data in Figure 3 are reasonably convincing in this regard, the data in Figure 4 are not. Without double staining with the same tetramer labeled with two different fluorochromes, it is harder to assess the validity of the staining in Figure 4. Except for 4B, the tetramer-binding cells are poorly resolved from the background. A control to show that no events are in the CD3^+^ CD8^+^ gate is absent. The data suggest that, for any antigenic peptide, equal numbers of precursors are available in the naive and the memory CD4 T cell compartments. These issues raise the concern that the detected cells are not binding the tetramers by their TCRs. About one third of the E641:I-A^b^-specific cells also bind to OVA323-339:I-A^b^ in young adults, not to mention old mice. This is an extraordinarily high number since peptides that bind to the same MHCII molecule need to share at least a couple of TCR contact amino acids to cross-react on the same TCR (Birnbaum et al., 2014). Since core nonamer sequences are not defined for either E641 or OVA323-339, it is not clear if they share any such TCR contact amino acids. The specificity concern is exacerbated by the fact that the cells that bind E641:I-A^b^ and OVA323-339:I-A^b^ do not undergo clonal expansion after immunization with E641. It is therefore essential to perform additional experiments with the resolving power of Figure 3 to provide convincing evidence that the tetramer binding being detected, particularly by ‘polyspecific’ CD4 T cells, is indeed via their TCRs.

3) It would be important to address directly the contribution of differences created during thymic selection versus those created during peripheral residence to the distinctions of naive CD4 T cell repertories between young and aged mice, such as data on repertoires of maturing thymocytes between young and aged mice, for example.

4) CD5 levels are interpreted throughout as indicators of self-pMHC affinity of TCRs. It would be important to have at least some data providing independent indications supporting this interpretation.

[Editors' note: further revisions were requested prior to acceptance, as described below.]

Thank you for sending your work entitled “Affinity for self drives the preferential accumulation of promiscuous CD4^+^ T cells over the lifespan” for consideration at *eLife*. Your revised manuscript was further evaluated by Tadatsugu Taniguchi (Senior Editor) and two reviewers, one of whom is a member of our Board of Reviewing Editors (BRE). Please find below the comments, which were assembled by the BRE member, and the offer we can make at this stage.

While the reviewers all find your work interesting, the revised manuscript does not substantively address all the four major concerns raised in the first review. Therefore, it does not cross the enthusiasm threshold for publication in *eLife* as a full article.

The first concern is that it would be useful to have some indication of the mechanisms by which TCR repertoire modifications during aging differ between CD4 and CD8 compartments. While the difficulty of providing this is acknowledged, it remains the case that the revised manuscript does not provide any additional data addressing such mechanistic possibilities, reducing cross-disciplinary impact of the manuscript. The second concern is that the seemingly high frequencies of polyreactivity the authors obtain in their analysis of CD4 T cell repertoires necessitate a stringent-ER identification of target-specific TCRs and/or some independent evidence (if not explanation) for high-frequency polyspecificity. This issue has been addressed, albeit mostly via plausible arguments rather than additional data. The third concern is regarding the thymic and/or peripheral origin of the reported repertoire differences. The authors have indeed provided very interesting data for this issue, and have directly and substantively addressed the concern, improving the significance of the findings in the manuscript. The fourth concern is regarding the use of CD5 levels as a surrogate marker for self pMHC-affinity of TCRs. While the revised manuscript makes a case for this by plausible arguments, the relationship still remains a hypothesis, albeit a reasonable one.

Under these circumstances, particularly in light of the lack of mechanistic insights, the scope of the impact of the manuscript is limited for its publication as a full *eLife* research article.

However, the findings in the manuscript concerning the re-shaping of the CD4 T cell repertoire in aging mice are novel and of great interest and, therefore, deserve rapid publication.

Under these circumstances, it is recommended that the manuscript be re-written as a ‘Short Report’. This is no doubt a challenge, since limits for a short report involve a 2000-word text limit and a four-item main display limit (http://elifesciences.org/category/short-report). Nonetheless, we hope that you will undertake this to get these interesting and provocative findings published in *eLife*.

---

## [Author Response]

*1) The manuscript demonstrates that the specificity (and perhaps self-pMHC affinity) landscape of the TCR repertoire shows interesting aging-related changes in the CD4 T cell compartment that are distinct from those reported for the CD8 T cell compartment (*[12]*). The findings are correlative at this point with no data implicating any mechanisms by which these distinctions arise, and they require greater rigor in both data and interpretation*.

We thank the reviewers for their constructive comments and critical input. Indeed, several studies have described a decrease in the numbers of CD8 T cells specific for foreign class I pMHC with aging. These include a decline in CD8 T cells specific for peptide epitopes from ovalbumin, vaccinia virus, influenza, HSV-1, LCMV (12; 38; 44; 54), and WNV (Janko Nikolich-Zugich, personal communication). In contrast, age-related changes in the capacity of the CD4 T cell compartment to bind foreign-pMHC have been largely unexplored. Our results show a significant increase in CD4 T cells that bind foreign-pMHC over time, even though the absolute number of CD4 T cells declines. Aging thus inversely impacts the CD4 and CD8 T cell compartments. We consider this to represent an important advance in our fundamental understanding of T cell biology. Although the mechanistic basis for these changes will benefit from further elaboration in future studies, here we establish a mechanistic link between tonic TCR interactions with self-pMHC, steady state proliferation in vivo (Figure 6 and 7), and the increase in binding capacity observed in this study (Figures 3 and 4). We have also now added analysis of thymocytes in response to major point #3 (Figure 5). These data support the notion that both thymic and peripheral pressures contribute to age-related changes in the binding capacity of the CD4 T cell compartment.

*2) The TCR specificity of the tetramer binding, the assay upon which all the results depend, must be established with greater rigor than at present. While the data in*
Figure 3
*are reasonably convincing in this regard, the data in*
Figure 4
*are not. Without double staining with the same tetramer labeled with two different fluorochromes, it is harder to assess the validity of the staining in*
Figure 4*. Except for 4B, the tetramer-binding cells are poorly resolved from the background. A control to show that no events are in the CD3*^*+*^
*CD8*^*+*^
*gate is absent. The data suggest that, for any antigenic peptide, equal numbers of precursors are available in the naive and the memory CD4 T cell compartments. These issues raise the concern that the detected cells are not binding the tetramers by their TCRs. About one third of the E641:I-A*^*b*^*-specific cells also bind to OVA323-339:I-A*^*b*^
*in young adults, not to mention old mice. This is an extraordinarily high number since peptides that bind to the same MHCII molecule need to share at least a couple of TCR contact amino acids to cross-react on the same TCR (Birnbaum et al., 2014). Since core nonamer sequences are not defined for either E641 or OVA323-339, it is not clear if they share any such TCR contact amino acids. The specificity concern is exacerbated by the fact that the cells that bind E641:I-A*^*b*^
*and OVA323-339:I-A*^*b*^
*do not undergo clonal expansion after immunization with E641. It is therefore essential to perform additional experiments with the resolving power of*
Figure 3
*to provide convincing evidence that the tetramer binding being detected, particularly by ‘polyspecific’ CD4 T cells, is indeed via their TCRs*.

We agree with the reviewers that tetramers require great care in both their use and data interpretation. With this in mind, our experiments were conducted with an extensive panel of dump antibodies as well as the two main published protocols for stringency in tetramer-based analysis: (i) dump tetramers (32; 40), or (ii) two-color tetramers (31; 36; 45). Each strategy has pros and cons. Dump tetramers gate out false-positive cells bound to some component of a tetramer – be it streptavidin (SA), the MHC, or the fluorescent protein (FP) used for the dump tetramer– in a TCR-independent manner. This method also excludes cells bound to two different tetramers (the dump and subject tetramers) in a TCR-specific manner, either because they express one TCR that binds both pMHC, or two TCRs that individually bind two distinct pMHC. The two-color approach selectively analyzes cells binding the same tetramer made with two different FPs on a diagonal, since tetramer binding is proportional to the affinity of TCR-pMHC interactions and the differently colored tetramers should bind a specific cell equivalently. This approach lowers the limit of detection for low affinity cells, since tetramers of both colors are competing for limited TCR space on a T cell, but it excludes from analysis those cells that bind exclusively to one of the two FPs as false-positives. Cells bound to SA, the MHC, or shared motifs between FPs (as could occur with the structurally related phycobiliproteins PE and APC) in a TCR-independent manner are included in the analysis as false-positives.

For Figure 4 we applied a sequential gating scheme (now Figure 4–figure supplement 5) to enumerate those cells that bind one tetramer exclusively (e.g. E641-only) after dumping out those cells that also bind the other tetramers (e.g. OVA and MCC) (Figure 4). Figure 4–figure supplement 6 has now been added to show how sequential gating impacts enumeration of the tetramer-bound populations under consideration. Theoretically, for the reasons outlined above, using two dump tetramers to enumerate the tetramer-only populations should be at least as stringent as the two-color approach.

A key point of emphasis is that we employed both the dump and two-color tetramer protocols to enumerate the ‘polyspecific’ CD4 T cells in question (e.g. E641+OVA). We realize from the Reviewer’s comments that our sequential gating scheme may not illustrate this point as well as a quadrant gate, and thus may cause confusion, even though it operationally achieves the same goal. Figure 4–figure supplements 8-10 have now been added to better illustrate the application of two-color staining, using quadrant gates, before and after the use of a dump tetramer (e.g. MCC) to identify cells binding strictly to two tetramers (e.g. E641+OVA). The numbers of ‘polyspecifc’ cells enumerated by two-color staining pre- and post-dump are also shown in these figure supplements. The derived numbers differ slightly from those obtained with our sequential gating scheme, shown in Figure 4, but the overall results and conclusions remain the same. We have also revised the text to better describe the logic behind our tetramer analysis (subsections “WNV-specific CD4^+^ T cells increase over time”, “Enumeration of CD4^+^ T cells binding multiple pMHC from the same mice” and “The CD4^+^ T cell compartment becomes more promiscuous for foreign pMHC over time”). Importantly, since the two-color approach is inherent within our analysis of the polyspecific cells in Figure 4, and applied after a dump tetramer, we consider the resolving power of these experiments to be at least equal to that of Figure 3.

We are also providing data to the reviewers from additional control experiments, which we performed with TCR negative hybridomas, in order to ensure that the stringency employed in Figure 4 provides resolving power equal to Figure 3. Here we used TCR negative 58α^-^β^-^ cells to model the two-color approach used in Figure 3, the dump approach used in Figure 4, and the use of both for polyspecific cells in Figure 4.

TCR negative 58α^-^β^-^ cells were incubated with E641 tetramers made with PE and APC, as in Figure 3, to model false-positive cells binding to both tetramers versus one or the other. A large number of events were collected since our tetramers are not generally “sticky” (Figure 5). We enumerated 222 false positive cells on the diagonal (of a total of 464 false-positives) per 500,000 cells collected. In the same experiment, 58α^-^β^-^ cells were incubated with E641-APC, OVA-PerCP-Cy5.5, and MCC-PE tetramers as in Figure 4. We enumerated 225 per 500,000 E641-bound cells without excluding cells binding OVA or MCC (Figure 5) and 126 per 500,000 E641-bound cells after excluded cells bound to OVA and MCC (Figure 5) as per the dump approach applied to E641-only cells in Figure 4. Both approaches reduce the number of false-positive cells enumerated in these experiments by a similar magnitude, suggesting to us that the analysis in Figures 3 and 4 are at least comparable in stringency.

Author response image 1.Exclusion of false-positive cells by two-color and dump tetramer analysis modeled with TCR negative T cell hybridoma. TCR negative 58α^-^β^-^ cells were incubated with E641: I-A^b^ tetramer in two color as in Figure 3 or with E641: I-A^b^, OVA: I-A^b^ and MCC: I-E^k^ tetramers with single color as in Figure 4. Contour plot showing our (A) live 58α^-^β^-^ cells; (B) 58α^-^β^-^ cells incubated with E641: I-A^b^ (APC) and (PE) for two-color analysis; or 58α^-^β^-^ cells staining positive for E641: I-A^b^ tetramer (C) pre and (D) post dump of cells bound to OVA: I-A^b^ and MCC: I-E^k^. Percent and number (of 500,000 58α^-^β^-^ cells) of cells bound to E641: I-A^b^ tetramer are shown in inset.**DOI:**
http://dx.doi.org/10.7554/eLife.05949.019

In Figure 6, we then combined the operating principles of both approaches to model our analysis of polyspecific cells that bind two distinct specificities (Figure 4). In Figure 6 we show two-color analysis of false-positive 58α^-^β^-^ cells binding E641, OVA, or both (39 of 500,000) after incubation with E641-APC, OVA-PerCP-Cy5.5, and MCC-PE tetramers as in Figure 4. Figure 6 then shows that the false positive cells binding both have largely been eliminated after those bound to MCC have been excluded as a dump (2 per 500,000). We interpret these data as evidence that combining both methods provides at least as much resolving power as that of Figure 3.

Author response image 2.Dump tetramer and two-color analysis excludes the bulk of false positives from the analysis. TCR negative 58α^-^β^-^ cells were incubated with E641: I-A^b^, OVA: I-A^b^ and MCC: I-E^k^ tetramers as in Figure 4. Contour plot showing 58α^-^β^-^ cells staining with E641: I-A^b^ (APC) and OVA: IA^b^ (PerCP- Cy5.5) pre and post using dump (MCC: I-E^k^) tetramer (A-B) Quadrant gate as in [31]. 500,000 58α^-^β^-^ events are displayed. Percent and number of within the quadrant gates are shown in inset.**DOI:**
http://dx.doi.org/10.7554/eLife.05949.020

Additional responses to Point #2 are as follows:

A) Figure 4–figure supplement 6 has been added to address the Reviewers’s comment that “About one third of the E641:I-A^b^-specific cells also bind to OVA323-339:I-A^b^ in young adults, not to mention old mice.” ∼5-7% of the total population of cells binding E641 also bind OVA in the adult or old mice if we use MCC as a dump. 14-20% bind OVA and E641 if we include triple binders. These could bind both pMHC with a single TCR. Or, this percentage is within the range of cells expressing two TCR alpha chains that could bind to two distinct tetramers via two distinct TCRs. Also, particularly without the dump tetramer, some of these cells are likely to be non-specific binders.

B) “A control to show that no events are in the CD3^+^ CD8^+^ gate is absent.”

Our anti-CD8 mAb was included in our panel of dump mAbs, as in other studies (36; 40), so it is difficult to confidently resolve CD8 T cells from the remainder of the cell populations we are dumping from our analysis. These were dumped since the literature suggests there should be some CD8 T cells binding class II pMHC, particularly the allo-pMHC, and we have no idea how age-associated changes in homeostatic or thymic pressures might impact MHC restriction (Hansen et al., 2013; [20]; Robey et al., 1991). We have added specificity data, shown in Figure 4–figure supplement 1, to complement the specificity controls already shown in the manuscript in order to further validate our reagents.

C) “…the same MHCII molecule need to share at least a couple of TCR contact amino acids to cross-react on the same TCR (Birnbaum et al., 2014).”

The reviewers make an important point that we have considered. Table 1 has now been added to show the E641 and OVA peptides with the predicted core nonamer sequences and putative TCR contact residues (P2, P5, and P8) (31; 56). Both peptides contain small hydrophobic side chains at P2 (Val and Ala, respectively) and polar residues at P5 (Asn and Glu, respectively). This suggests some weak similarities between TCR contact residues of the two peptides that may hint at a mechanistic basis for the cross reactivity we observe.

The study cited above does not rule out the possibility of polyspecific cells that bind two distinct, or at least more distantly related pMHC in a TCR-dependent manner. Indeed, they pointed out that examples exist in the literature for TCRs recognizing non-homologous peptides (Birnbaum et al., 2014). Furthermore, TCRs have been described that focus more on the MHC than the peptide and even cross-react with class I MHC after selection with a single class II pMHC (20). In addition, Jenkins and colleagues showed CD4 SP thymocytes binding both 2W:I-A^b^ and IgM:I-A^b^ or 2W:I-A^b^ and FliC:I-A^b^ tetramers under conditions of limiting negative selection (8). By our assessment (not shown), the putative core nonamers of the former share hydrophobic residues at P2 (Trp and Val) and P5 (Leu and Ala) that vary in size, as do the latter at P5 (Leu and Ile) and P8 (Trp and Leu). We think it is therefore possible that exceptions to the requirement of shared TCR contacts for cross reactivity become more frequent under conditions of limiting negative selection with aging. Also, since ∼10% of peripheral T cells express dual TCRs (which represent > 40% of alloreactive cells), the possibility exists that our polyspecific cells may be recognizing multiple pMHC via dual TCRs (33). We think this is an important and exciting future direction that extends beyond what can be reasonably explored in one study.

*3) It would be important to address directly the contribution of differences created during thymic selection versus those created during peripheral residence to the distinctions of naive CD4 T cell repertories between young and aged mice, such as data on repertoires of maturing thymocytes between young and aged mice, for example*.

We agree that this line of inquiry is important for a deeper mechanistic understanding of our findings. To address this comment, we first analyzed the relationship between CD5 expression on adult and old peripheral T cells versus CD4 SP thymocytes (now shown in Figure 1–figure supplement 1). CD5 levels are higher on both adult and old CD4 SP thymocytes than on total adult CD4 T cells, but not different from each other. No clear connection can be made with these data and our previous results.

We also performed anti-His tag-based tetramer enrichment of adult and old thymocytes with E641, OVA, and MCC tetramers (now shown in Figure 5 and Figure 5–figure supplement 1). In consideration of the Reviewers’ comments, each tetramer had a unique color (E641 = PerCpCy5.5; OVA = PE-Cy7; MCC = PE) and all tetramers also had a common color (APC). Here we dumped the unique colors of two of the tetramers (e.g. OVA and MCC) and then enumerated those cells that bind the remaining tetramer in two colors (e.g. E641). Since old mice had ten-fold fewer (∼2X10^7^) total thymocytes than adult mice (∼2X10^8^), thymocytes from 4-5 old mice were pooled for tetramer enrichment. This was repeated with two data sets representing 4 mice/group and one representing 5/group from a total of three experiments. For the adults in each experiment, an aliquot of 2X10^7^ cells from individual mice were pooled to equalize our tetramer enrichment samples between the adult and old populations.

More E641-binding CD4 SP thymocytes were enumerated from equivalent numbers of thymocytes from old mice compared to adults, indicating that the percentage of E641-bound CD4 SP is higher in old mice. The number of tetramer bound cells for OVA and MCC were not significantly different between adult and old in our sample size. The numbers of tetramer bound cells were so low that we are not surprised we did not detect cells binding two distinct tetramers, given their relatively low frequency in the periphery. Since Fink and colleagues have reported that thymic output is consistent over time relative to thymic size (19), the implication is that on a daily basis a higher number of E641 binders leave the thymus in old mice compared with the adult to take up residence in the periphery. Clearly, this is an important direction for further elaboration in future studies.

*4) CD5 levels are interpreted throughout as indicators of self-pMHC affinity of TCRs. It would be important to have at least some data providing independent indications supporting this interpretation*.

We agree with the reviewers on this point as this was the motivation for the data shown in Figure 1 and Figure 2. First we assessed relative TCR levels, since TCR down regulation occurs upon TCR engagement, and demonstrated that they are inversely correlated with relative CD5 levels (Figure 1). We also assessed BrdU incorporation, which is dependent on TCR engagement, and showed higher BrdU incorporation occurred in cells with higher CD5 levels (Figure 2). The data in these figures were presented to complement the studies we referenced in the text (2; 26; 27; 37; 43). Such work is also consistent with a recent study from Jameson and colleagues (16). We think the sum of the data in these bodies of work provide sufficient justification for our use of CD5 expression as a surrogate measure of tonic TCR interactions for self-pMHC.

[Editors' note: further revisions were requested prior to acceptance, as described below.]

*[…] While the reviewers all find your work interesting, the revised manuscript does not substantively address all the four major concerns raised in the first review. Therefore, it does not cross the enthusiasm threshold for publication in* eLife *as a full article. […]Under these circumstances, it is recommended that the manuscript be re-written as a ‘Short Report’*.

We thank all involved individuals for their time and effort spent evaluating our work, as well as your offer to condense our key findings into a Short Report. Our manuscript now contains the following:

Figure 1: The broad characterization of naive and memory phenotype CD4 T cells regarding absolute numbers, CD5 expression, TCR down regulation, and steady-state proliferation (BrdU) (previously Figures 1 and 2). These data show that the CD4 T cell repertoire contracts in absolute number with aging, shifts in composition towards cells with a memory phenotype, and accumulates an increased frequency of constituents with higher CD5 levels that proliferate.

Figure 2: The enumeration of tetramer specific cells that bind only one tetramer, and those that bind two tetramers (previously Figures 3 and 4). These data demonstrate that a higher number of CD4 T cells bind to one or two individual pMHC within the contracting CD4 T cell repertoire of old mice compared with adults.

Figure 3: The enumeration of tetramer specific thymocytes. These data provide evidence that age-related changes in thymic selection serve as a potential mechanistic basis for changes in peripheral binding capacity (previously Figure 5).

Figure 4: Present the link between CD5 expression (as a measure of affinity for tonic TCR-pMHC interactions) and fold expansion of tetramer-bound cells over time (previously Figure 6E and 6F), as well as the relationship between CD5 levels and steady state proliferation (previously Figure 7). These data provide evidence that peripheral pressures (i.e. homeostatic proliferation driven by tonic TCR-pMHC interactions) serve as a potential mechanistic basis for changes in peripheral binding capacity that re-shape the aging CD4 T cell repertoire.